# Effects of Dietary Supplementation of L-Carnitine and Excess Lysine-Methionine on Growth Performance, Carcass Characteristics, and Immunity Markers of Broiler Chicken

**DOI:** 10.3390/ani9060362

**Published:** 2019-06-16

**Authors:** Seyed Mohammad Ghoreyshi, Besma Omri, Raja Chalghoumi, Mehrdad Bouyeh, Alireza Seidavi, Mohammad Dadashbeiki, Massimo Lucarini, Alessandra Durazzo, Rene van den Hoven, Antonello Santini

**Affiliations:** 1Department of Animal Science, Rasht Branch, Islamic Azad University, Rasht 41335-3516, Iran; sm.ghoreyshi@chmail.ir (S.M.G.); booyeh@iaurasht.ac.ir (M.B.); 2Laboratory of Improvement and Integrated Development of Animal Productivity and Food Resources, Department of Animal Science, College of Agriculture of Mateur, University of Carthage, Bizerte 7000, Tunisia; omribesma1@gmail.com (B.O.); chalghoumi.r@hotmail.com (R.C.); 3Department of Veterinary Science, Rasht Branch, Islamic Azad University, Rasht 41335-3516, Iran; dadashbeigi@gmail.com; 4CREA-Research Centre for Food and Nutrition, Via Ardeatina 546, 00178 Rome, Italy; massimo.lucarini@crea.gov.it (M.L.); alessandra.durazzo@crea.gov.it (A.D.); 5Clinical Unit of Equine Internal Medicine, Veterinarmedizinische Universitat, 1210 Wien, Austria; rene.vandenhoven@vetmeduni.ac.at; 6Department of Pharmacy, University of Napoli Federico II, 80138 Napoli, Italy

**Keywords:** amino acids, dietary supplementation, broiler, growth performance, humoral immunity

## Abstract

L-carnitine as well as lysine and methionine are amino acids of important nutritional and nutraceutical interest and are used in nutritional strategies as diet supplements to improve feed quality characteristics in animals and broiler chicken in particular. This study investigated the effect of different levels of L-carnitine and extra levels of lysine-methionine on growth performance, carcass characteristics, and some immune system markers. Two hundred seventy male Ross 308 broilers were a fed control diet (C) and eight different diets supplemented with an excess of amino acids. In the experimental diets, identified as D1, D2, D3, D4, D5, D6, D7, and D8, extra L-carnitine, lysine, and methionine were added in excess with respect to the American National Research Council (NRC) recommendations: L-carnitine equal to NRC (D1); control diet supplemented with lysine at 30% in excess of NRC, methionine at 30% in excess of NRC, and L-carnitine equal to NRC (D2); control diet supplemented with lysine equal to NRC, methionine equal to NRC, and L-carnitine at 15% in excess of NRC (D3); control diet supplemented control diet supplemented with lysine at 15% in excess of NRC, methionine at 15% in excess of NRC, and L-carnitine at 15% in excess of NRC (D4); control diet supplemented lysine at 30% in excess of NRC, methionine at 30% in excess of NRC, and L-carnitine at 15% in excess of NRC (D5); control diet supplemented with lysine equal to NRC recommendations, methionine equal to NRC recommendations, and L-carnitine at 75% in excess of NRC (D6); control diet supplemented with lysine at 15% in excess of NRC, methionine at 15% in excess of NRC, and L-carnitine at 75% in excess of NRC (D7); and control diet supplemented with lysine at 30% in excess of NRC, methionine at 30% in excess of NRC, and L-carnitine at 75% in excess of NRC (D8). During the starter and growth phases, feed intake was not affected by dietary treatment (*p* > 0.05). By contrast, body weight and FCR were both affected (*p* < 0.001) during the starter period. During the finisher phase, feed consumption was affected (*p* < 0.05) by dietary treatment. Feed intake of broilers fed on C, D3, D6, and D7 were statistically similar (*p* > 0.05) (1851.90, 1862.00, 1945.10, and 1872.80 g/pen/day, respectively) and were higher (*p* < 0.05) than 1564.40 g/pen/day (D5). With the exception of drumsticks, neck, back thoracic vertebrae, and proventriculus weights, economical carcass segments were not affected (*p* > 0.05) by the dietary supplementation of amino acids. Duodenum and ileum weights and lengths decreased with amino acid supplementation (*p* < 0.05). IgT and IgG titers against Sheep Red Blood Cells (SRBC) for both primary and secondary responses were not affected by dietary treatments (*p* > 0.05). Dietary amino acids supplementation did not affect IgM titer after the secondary challenge (*p* > 0.05) and had a significant effect (*p* < 0.05) on serum antibody titers in broilers vaccinated against Newcastle disease (NCD) and Gumboro ‘s disease at the 27th and 30th days, respectively.

## 1. Introduction

Nowadays, the growing demand for poultry meat has resulted in pressure on breeders to increase the growth rate of birds, the feed efficiency, the size of breast muscles, and the reduction in abdominal fatness [1]. Therefore, research is being oriented toward improving the techniques of poultry meat production. The improvement in carcass compositions with additives has become a focus on nutrition research. As an example, the addition of amino acids and metabolic intermediates to diets may lower the abdominal fat deposition in poultry. One example is L-carnitine, the biological active form of carnitine, which is synthesized in the liver, kidney, and brain [2] from the essential amino acids lysine and methionine, that can be considered as L-carnitine precursors [3,4]. L-carnitine (δ-trimethylamino-β-hydroxybutyrate) is a quaternary hydrosoluble amine with a small molecular weight that occurs naturally in microorganisms, plants, and animals [5]. Its concentration in animals varies according to species [6], tissue type [4,7], nutritional status of the animal [8,9], and the feed quality [10]. Dietary effects of L-carnitine, lysine, and methionine supplementations on the growth performance and body composition of broiler chickens are still poorly understood. Many researches have suggested that the dietary addition of lysine and methionine in excess with respect to the American National Research Council NRC [11] recommendations may result in enhanced performances, especially with regard to breast meat yield, body weight gain, and feed conversion ratio [12,13,14,15,16,17,18,19,20,21]. Corzo et al. [22] reported how a high dietary density of amino acids can lead to an increased breast meat related to an increase of lean muscle tissue. Moreover, Mukhtar et al. [10] reported that a significant improvement in amino acids feed intake improves the average body weight gain and feed conversion ratio. On the other hand, Si et al. [15] concluded that the level of methionine should not be increased if lysine is in excess of its minimum needs.

It has been also reported that the dietary supplementation of lysine and methionine can improve the immunity of broiler chickens against different diseases [23,24,25,26]. Moreover, it has been reported that methionine constructively affects the immune system by improving both cellular and humoral responses [27,28,29]. The mechanisms proposed to explain methionine interference in the immune system is the T cells proliferation, which are sensitive to intracellular glutathione and cysteine levels, compounds that participate in the methionine metabolism [30].

L-carnitine supplementation is used to improve broiler productivity [31], and its bioavailability depends on the composition of the diet. Theoretically, supplementing broiler diets with an adequate content of L-carnitine would facilitate the fatty acids β-oxidation and decrease the esterification reactions and triacylglycerols storage in the adipose tissue [3,32,33]. However, the impact of extra supplied L-carnitine may depend on the magnitude of its endogenic biosynthesis from lysine and methionine in the presence of Fe^2+^ and a number of vitamins (e.g., ascorbate, niacin, and pyridoxine), which are required as cofactors for the enzymes involved in the metabolic pathway of L-carnitine [4,34,35,36]. Some authors reported that abdominal fat deposition in broilers is reduced by L-carnitine supplementation without a significant effect on daily gain or feed conversion [37], while others observed no impact of dietary L-carnitine supplementation on abdominal fat composition [38]. Nonetheless, Bouyeh and Gevorgyan [39] and Celik et al. [40] showed that the growth performance of broilers was improved by L-carnitine supplementation. A study by Hosseintabar et al. [33] evaluated the effects of different levels of L-carnitine, lysine, and methionine on the blood concentrations of energy, protein, and lipid metabolites of male broiler chickens and concluded that, compared to a standard diet, the addition of 150 mg/kg of L-carnitine plus 15% lysine and methionine sustained a low plasmatic total cholesterol concentration compared to a standard diet. El-Wahab et al. [41] reported the broilers fed high levels of lysine and methionine with a surplus amount of L-carnitine (350 mg kg^−1^ to the diet) led to significantly lower cholesterol levels vs. a low L-carnitine intake. To the knowledge of the authors, the effect of simultaneous feed supplementation with L-carnitine and excess lysine-methionine on growth performance, carcass characteristics. and immunity markers of broiler chicken has not been studied before.

There is a need to standardize the dose of lysine-methionine and L-carnitine supplementation in the diet of broiler chickens not only to enhance their growth performance and carcass characteristics but also to improve their immune response. Hence, the main objective of this study has been to evaluate the effect of dietary supplementation of different levels of L-carnitine with or without an excess of lysine-methionine compared to dietary nutrient requirements on broiler chickens’ growth performance, humoral immunity markers, and carcass characteristics during a 6-week rearing trial.

## 2. Materials and Methods

### 2.1. Animal Welfare and Ethics

All procedures related to animals’ care and sampling were conducted under the approval of the Institution’s Ethic Committee at the Department of Animal Science, Rasht Branch, Islamic Azad University, Rasht (Iran) (protocol N° 105/19) before the beginning of the experimental trial.

### 2.2. Experimental Diets Preparation

A control diet (C) for broiler chicken based on corn and soybean-meal was prepared according to the dietary nutrient requirements of broilers [1]. The ingredients and chemical composition of the control diet are given in Table 1. Thereafter, eight amino acids-supplemented diets (indicated as D1, D2, D3, D4, D5, D6, D7, and D8) were prepared by mixing the control diet thoroughly with the designated supplements at the required incorporation levels as shown in Table 2.

As stated above, the L-carnitine, lysine, and methionine levels in starter, grower, and finisher feeds for the control diet were determined according to the NRC [11] recommendations. In the experimental diets, D1, D2, D3, D4, D5, D6, D7, and D8, extra L-carnitine, lysine, and methionine (Carniking1, Lonza Ltd., Basel, Switzerland) were added in excess to the NRC [11] recommendations, as shown in Table 2, since NRC [11] recommended diets are suggested for feeding Ross 308 broiler chickens because of fewer phases of feeding periods and lower workloads [42].

### 2.3. Animals and Experimental Design

Two hundred seventy 1-day-old, male Ross 308 broiler chicks obtained from a local commercial hatchery were used in this experiment. Chicks were randomly distributed into 27 pens (9 groups × 3 replications, each replication included 10 chicks). Each group was allocated to one of the 9 dietary treatments indicated above. Birds were given starter feed from 1 to 21 days, a grower feed from 22 to 35 days, and a finisher feed from 36 to 42 days of age. Feed and water were provided ad libitum throughout the experimental assay. For the growth performance traits, the experimental unit was the pen. For the carcass and immunity traits, the experimental unit was the chicken.

### 2.4. Growth Performance Monitoring

Tens birds per pen were weighed together on the 1^st^, 21^st^, and the 35^th^ days of age to determine the live body weight and the weight gain. The feed consumption and feed conversion ratio (FCR) were also calculated for each growing phase as follows: from the 1^st^ to the 21^st^ day, from the 22^nd^ day to the 35^th^ day, and from the 36^th^ to the 42^nd^ day of the 42 days experimental study as described by Bouyeh and Gevorgyan [39].

### 2.5. Carcass Characteristics Determination

As shown by Panda et al. [43,44,45], at the end of the experiment (day 42), three broilers per same treatment (e.g., one broiler per same diet per pens) (*n* = 3) were randomly selected, weighed without prior fasting, and scarified between 9:00 am and 10:00 am by cervical dislocation to evaluate the characteristics of the carcass. After skin removal and total evisceration, the feet were separated from the carcass in the tibio–tarsal joint. Economic carcass and gastrointestinal segments were removed, weighed, and the ratios of each segment to body weight were calculated.

### 2.6. Humoral Immune Response Measurements

Non-pathogenic antigens of Sheep Red Blood Cells (SRBC) were used to monitor the humoral immune response of broilers. The SRBC were purchased from a local Iranian supplier. A suspension was prepared by mixing 1 mL of phosphate-buffered saline (PBS) with 10 mL of SRBC. Six birds per same treatment (e.g., two broiler per same diet per pens) (*n* = 6) were subcutaneously injected in the breast with 0.5 mL of SRBC suspension on the 22^nd^ and the 36^th^ days of the experimental trial.

Then, 7 days after each sensitization (28 and 42 days, respectively), antibody titers against SRBC were measured by a hemagglutination inhibition (HI) test according to Cunningham [46]. All antibody titers were recorded according to previous studies [47,48].

Birds were also vaccinated against infectious bronchitis (IB) on the 1^st^ and 16^th^ days of age, against Newcastle disease (NCD) on the 8^th^ and 20^th^ days of age, and against Gumboro’s disease on the 14^th^ and 23^rd^ days of age. The humoral immune responses of chickens to the IB virus at the 23^rd^ day of age, to the NCD virus at 27^th^ day of age, and to the Gumboro virus at the 30th day of age were measured using the HI and ELISA methods as described by references [47,48]. Blood samples were collected from the brachial vein. Serum was separated by centrifugation (3000× *g* rpm for 15 min), and antibody titers against IB, NCD, and Gumboro virus were measured using commercially available ELISA kits (Bio-check BV, Gouda, Holland) according to the manufacturer’s instructions. The absorbance of controls and samples were read at a wavelength of 405 nm using an ELISA reader (Bio-Tek Instruments Inc. ELX 800; Winooski, VT, USA).

### 2.7. Statistical Analysis

All data were subjected to an ANOVA statistical analysis with the General Linear Model (GLM) procedure of SAS [49]. The GLM was used according to the following model:Y_ijk_=µ+α_j_+β_k_+(αβ)_jk_+ɛ_ijk_(1)
where Y_ijk_ = the j^th^ observation on the i^th^ treatment, µ = overall mean, α_j_ = the main effect of the L-carnitine level, β_K_ = the main effect of the methionine-lysine level, αβ_jk_ = The effect of the interaction of L-carnitine and of methionine-lysine treatments, and ɛ_ijk_ = The random error

## 3. Results

### 3.1. Growth Performance

Table 3 summarizes the results of the growth performance using the different diets. During the starter and growth phases, feed intake was not affected by dietary treatment (*p* > 0.05). By contrast, body weight and FCR were both affected (*p* < 0.001) during the starter period. In fact, D1, D2, D3, D6, D7, and D8 were associated with the highest live body weight (*p* < 0.001) with mean values of 706.31, 745.50, 671.10, 733.67, 741.00, and 723.27 g/pen/21day, and consequently, the feed conversion ratio was the lowest (*p* < 0.001) during this period compared to the Control Diet (C).

However, during the grower period, the live body weight of the broiler fed on D1 and D2 was not different from that of C (1241.29 g/pen/period and 1165.50 g/pen/period vs. 1207.10 g/pen/period) and was higher than those of D3, D4, D5, D6, D7, and D8 (*p* < 0.001). D1 and D2 were associated with the lowest feed conversion ratio (*p* < 0.001) with mean values of 2.33 and 2.48, respectively. During the finisher phase, feed consumption was affected (*p* < 0.05) by dietary treatment. Feed intake of broilers fed on C, D3, D6, and D7 were statistically similar (*p* > 0.05) (1851.90, 1862.00, 1945.10, and 1872.80 g/pen/d, respectively) and were higher (*p* < 0.05) than 1564.40 g/pen/day (D5).

However, live body weights were similar between dietary treatments (*p* > 0.05) with mean values of 1976.53 g/pen/period and 1949.27 g/pen/period, respectively. The FCR of this period was not affected (*p* > 0.05) by dietary treatment.

### 3.2. Carcass Characteristics

The results obtained for carcass parameters (economical carcass segments, body organ segments, and gut organs) are shown in Table 4.

Our results indicate that, with the exception of thigh weights, economical carcass segments were not affected by dietary treatment (*p* > 0.05). The neck weights of the bird given D1 and D8 were the highest (*p* > 0.05) with mean values of 64.50 g and 64.63 g, respectively, compared to 79.57 for the control diet. D2 was associated with the highest back thoracic vertebrae weight (*p* > 0.05) at 103.99 g versus 45.11 g for birds given the diet D8. However, all other body segments (heart, liver, gizzard, and abdominal fat) weights were not affected by dietary treatment (*p* > 0.05). With the exception of proventriculus weight, gut organ weights were not affected (*p* < 0.05) by dietary treatment. Our data showed that the dietary supplementation of lysine and methionine at 30% with a level of L-carnitine equal to NRC and of 15% of lysine, methionine, and carnitine increased (*p* < 0.05) proventriculus weight from 10.77 g to 11.26 and 11.43 g, respectively.

### 3.3. Intestine Segments

The effects of dietary treatment on the length, weight, width, and diameter of small intestine segments (duodenum, jejunum, and ileum) in broilers are shown in Table 5.

The dietary supplementation of amino acid decreased (*p* < 0.05) duodenum weight and length. Birds given a supplemented diet with 30% of lysine and methionine in excess of NRC and 75% of L-carnitine in excess of NRC had the lowest weight, with a mean value of 12.99 g. The lowest duodenum length was recorded when birds were given 15% of lysine and methionine in excess of NRC plus 75% of L-carnitine in excess of NRC. A combination of L-carnitine (75%) and lysine-methionine (30%) increased the duodenum width significantly (*p* < 0.05) from 7.50 mm to 9.38 mm. Concerning the duodenum diameter, the dietary addition of lysine-methionine at a level equal to NRC plus 15% of L-carnitine in excess of NRC (D3), of lysine-methionine and L-carnitine at a rate of 15% in excess of NRC (D4), and of lysine-methionine and L-carnitine at a rate of 30% in excess of NRC plus 15% of L-carnitine in excess of NRC (D5) increased the diameter from 0.94 mm (C) to, respectively, 1.61mm, 1.67 mm, and 1.68 mm.

Dietary treatments did not affect (*p* > 0.05) the jejunum weight and length. The dietary addition of lysine-methionine at a level equal to NRC plus 15% of L-carnitine in excess of NRC (D3) and of lysine-methionine and L-carnitine at a rate of 15% in excess of NRC (D4) increased the jejunum width (*p* < 0.05) from 9.26mm (C) to 9.50 mm (D3) and 9.80 mm (D4). The dietary supplementation of amino acids significantly decreased (*p* < 0.05) the ileum weight from 9.10 g (C) to 2.67 g (D4) and 2.97 g (D7) and the length from 18.16mm to 9.26 mm (D8) and 8.00 mm (D7). The ileum width of the broiler group fed with lysine-methionine at a level of 15% plus L-carnitine at 75% in excess of NRC was the highest (*p* < 0.05) compared to the control diet 7.33 mm versus 7.97 mm (D7). Dietary treatment did not affect (*p* > 0.05) ileum diameter.

Caecum weights were not different between treatments and ranged from 12.62 g (D5) to 17.13 g (C) (*p* > 0.05). The colon weight of broiler bird fed with lysine-methionine at a level equal to NRC plus 15% of L-carnitine in excess of NRC (D4) was the highest (*p* < 0.05), with a mean value of 2.36 g. Rectum weights were similar between all the treatments groups and ranged from 1.78 g (D6) to 2.32 g (D7) (*p* > 0.05).

### 3.4. Humoral Immune Response

#### 3.4.1. Humoral Immune Response Against Sheep Red Blood Cell (SRBC)

The dietary effect of amino acids supplementation on primary and secondary antibody responses are shown in the Table 6. IgT and IgG titers against SRBC for both primary and secondary responses were not affected by dietary treatments (*p* > 0.05). Birds receiving diets supplemented with excesses of amino acids (D1, D2, D3, D4, D5, D6, D7, and D8) had significantly lower titers of IgM than that of those receiving C for primary response (*p* < 0.05), with mean values of 2.33 log_10_ versus 2.00 log_10_ (D1); 1.66 log_10_ (D2); 1.33 log_10_ (D3); 1.33 log_10_ (D4); 1.00 log_10_ (D5); 1.33 log_10_ (D6); 1.33 log_10_ (D7); and 2.00 log_10_ (D8). Dietary treatment did not affect (*p* > 0.05) IgM titer after a secondary challenge.

#### 3.4.2. Humoral Immune Response Against Bronchitis, Newcastle and Gumboro diseases

The influence of different amino acids dietary supplementations on serum antibody titer in chickens vaccinated against IB, NCD, and Gumboro virus are shown in Table 7. Serum antibody titers in broilers vaccinated against bronchitis were not affected (*p* > 0.05) by dietary treatment on day 23. However, a significant increase (*p* < 0.05) in serum antibody titers in broilers vaccinated against NCD and Gumboro on the 27th and the 30th days was observed from, respectively, 3.66 log_10_ (C) to 6.00 log_10_ (D3) and from 3.38 log_10_ (C) to 3.59 log_10_ (D6). Chickens fed with 30% of lysine-methionine in excess of the NRC and of L-carnitine equal to the NRC supplemented diet had the heaviest thymus (*p* < 0.001) with a mean value of 24.60 g versus 15.59 for the control diet. However, chickens fed on L-carnitine (15%) and lysine-methionine (15%) in excess of the NRC had the lightest thymus with a mean value of 7.17 g. Bursa of fabricius and spleen weights were not affected (*p* > 0.05) by dietary L-carnitine and lysine-methionine supplementation.

## 4. Discussion

It has been observed that L-carnitine supplementation can enhance broiler productivity [26]. However, the effect of administrating an excess of L-carnitine may depend on the magnitude of its endogenic biosynthesis from lysine and methionine [20,29,30,31]. In the current study, we are reporting the effect of dietary supplementation of different levels of L-carnitine with or without an excess of lysine-methionine on growth performance, carcass traits, and some humoral immunity markers of broiler chicken.

### 4.1. Growth Performance

The results relative to the growth performance show that a combination of lysine and methionine at 15% and 30% in excess of the NRC recommendations was without impact on the feed intake, live body weight gain, and FCR of broiler chicken during the whole experiment period.

Live body weights at the end of the experimental trail varied from 2828.60 g (D2) to 3602.70 g (D4). These values are consistent with those reported by Bouyeh and Gevorgyan [39], who showed that dietary supplementations of lysine-methionine at 0, 1.1, 1.2, 1.3, or 1.4% higher than NRC recommendations [11] led to body weight gains from mean values of 2960 g to 2920 g (1.1%), 2850 g (1.2%), 2970 g (1.3%), and 2730 g (1.4%), respectively.

On the other hand, Hickling et al. [12] showed that broilers fed with diets supplemented with methionine at a level as suggested by NRC and at a level 112% in excess with respect to the NRC recommendations weighed at 6 weeks 2221 g and 2248 g, respectively, and had feed conversion efficiencies of 1.81 (NRC) and 1.79 (112% of methionine in excess of the NRC), respectively. Birds fed with four levels of lysine—equal to the NRC recommendations and 106%, 112%, and 118% in excess of the NRC recommendations—weighed 2221 g, 2227 g, 2234 g, and 2238 g, respectively, at 6 weeks of age and had feed conversion efficiencies of 1.81 (NRC), 1.81 (106% of lysine in excess of the NRC), 1.80 (112% of lysine in excess of the NRC), and 1.79 (118% of lysine in excess of the NRC). Mukhtar et al. [50] found that dietary increasing of lysine from 53% to 78% and of methionine from 36% (control diet) to 61% had a significant effect on feed intake, body weight gain, and feed conversion. Mukhtar et al. [50] also reported a significant improvement of feed intake, average body weight gain, and feed conversion ratio when broiler chicken were fed with five diet: diet A (1.2% lysine + 0.49 methionine) without a broiler supper concentrate, used as control; diet B similar to diet A but with a broiler supper concentrate; diet C (1.3 lysine+ 0.56 methionine); diet (D) (1.4 lysine + 0.6%methionine); and diet (E) (1.5% lysine + 0.63% methionine). Body weight gain increased from 1080.76 g (A) to 1806.75 g (B), 1828.31g (C), 1834.93 g (D), and 1940.0 g (E). Feed conversion ratio decreased from 2.32 (A) to 1.97 (B), 1.95 (C), 1.94 (D), and 1.93 (E). More recently, Bouyeh and Gevorgyan [51] reported that the dietary incorporation of lysine and methionine at 0, 10%, 20%, 30%, and 40% in excess of the NRC recommendations [11] did not affect the body weight gain at 42 days of age but that the feed conversion ratio was affected by dietary treatment at 21 days of age.

With regard to the L-carnitine dietary supplementation, our results show that a dietary inclusion of L-carnitine at 75% or 15% in excess of NRC recommendations increased the live body weight, feed intake, and FCR of 308 Ross broilers. In this respect, our data are in line with those obtained by Celik et al. [40]. These authors reported that the growth performance of broilers was improved by L-carnitine supplementation at a level of 50 mg/l in drinking water.

As far as L-carnitine was concerned, the results reported here are consistent with those reported by Rodehutscord et al. [52] and by Farrokhyan et al. [32], although in both studies, carnitine supply was not supplied alone but in combination with other nutrients or additives. Indeed, Rodehutscord et al. [52] studied the effect of adding 80 mg of L-carnitine per kg of diet with two dietary levels of fat (namely 4 and 8%) on growth performance of broiler chickens. At the end of the trial, on day 21, the live body weight averaged 853 g and feed conversion was improved by almost 5% in chicken groups receiving L-carnitine supplemented diets. Farrokhyan et al. [32] also examined the effect of dietary combinations of 0, 150, and 300 mg/kg of L-carnitine with or without 1 g/kg or 2 g/kg of gemfibrozil, on the growth performance of broilers. It has been observed that, as dietary L-carnitine increased, weight gain and birds’ feed intake increased and FCR decreased. However, our results are not in agreement with those of other studies. Barker and Shell [53] showed that the dietary addition of L-carnitine at 0.50 or 100 mg/kg diet did not affect the weight gain or feed efficiency of broiler chicken. Lien and Horng [54] demonstrated that diets supplemented with 160 mg L-carnitine/kg for 6 weeks did not affect broilers’ feed intake, body weight gain, and feed conversion ratio. Corduk et al. [55] also reported that the dietary addition of L-carnitine at 100 mg/kg did not influence body weight gain, feed intake, and feed conversion ratio of broiler chickens.

Our data show that the dietary combination of L-carnitine, lysine, and methionine had a significant effect on growth performance. In fact, L-carnitine is an amine compound biosynthesized primarily in the liver from the amino acids lysine and methionine. It is involved in energy metabolism, where it is required for the transport of long-chain fatty acids into the mitochondrial matrix for β-oxidation by the fatty acid oxidation complex [3]. One the other hand, Murray et al. [56] found that the addition of synthetic amino acids like lysine and methionine at high levels to the diet can stimulate insulin secretion from pancreas by aggregating in plasma which in turn releases amino acids and fatty acids [57] from the bodily saved sources and leads to protein synthesis.

Adding 75% of L-carnitine plus 15% or 30% of lysine-methionine or 15% of L-carnitine plus 15% of lysine-methionine had a significant effect on feed intake and live body weight at the end of the experimental trial. Furthermore, adding 15% of L-carnitine plus 30% of lysine-methionine had no benefit and actually reduced feed intake and body weight and increased FCR significantly compared to the control.

To our knowledge, no information related to the effect of supplementing L-carnitine in combination with lysine-methionine on broiler growth performance is reported in the literature until now.

### 4.2. Carcass Characteristics

With the exception of drumsticks, neck, back thoracic vertebrae, and proventriculus weights, all other economical carcass segments weights were not affected by the dietary supplementation of L-carnitine and lysine-methionine. With respect to lysine-methionine supplementation, Mukhtar et al. [50] studied the effect of lysine and methionine on broilers’ carcass characteristics and reported that the dietary inclusion of A (1.2% lysine + 0.49% methionine) without broiler supper concentrate, used as control; B similar to diet A but with a broiler supper concentrate; C (1.3% lysine + 0.56 %methionine); D (1.4% lysine + 0.6% methionine); and E (1.5% lysine + 0.63% methionine) increased the eviscerated carcass weight from 1036.46 (A) to 1761.75 (B), 1783.21(C), 1790.93 (D), and 1894.6 (E) and the yield of commercial cuts (breast and drumstick). The percentage of meat in the drumsticks increased from 72.17% (A) to 76.54% (B), 81.76% (C), 78.67% (D), and 82.35% (E). Concerning the percentage of meat of the breast, it increased from 77.01% (A) to 80.33 (B), 85.84 (C), 86.55 (D), and 86.89 (E). However, Bouyeh and Gevorgyan [39] found that the dietary supplementation of lysine-methionine at levels of 0, 10, 20, 30, or 40% more than the NRC [11] recommendation did not affect the thigh and leg percentage to carcass weight. However, it had a significant effect on breast meat yield, carcass traits, and abdominal fat pad, and liver and heart weights. Concerning the effect of L-carnitine supplementation on carcass traits, previous studies have shown that the dietary inclusion of L-carnitine did not affect abdominal fat, heart, and liver weights [36,53,54,58]. On the other hand, Farrokhyan et al. [32] found that the dietary supplementation of L-carnitine (300 mg/kg) reduced abdominal empty carcass from 1826.6 g/chiks to 1793.3 g/chiks and breast weight from 1566.6 g/chiks to 1563.3 g/chiks. This dietary supplementation did not affect wing weight. The limited effect of L-carnitine observed in the present study could be attributed to a limited intestinal absorptive capacity of L-carnitine. Another possible explanation is that L-carnitine is easily degraded by intestinal microflora as suggested by Xu et al. [37].

### 4.3. Intestine Segments

In the present study, the dietary supplementation of lysine-methionine and L-carnitine had a significant effect on all intestine segments with the exception of jejunum weight and length, ileum diameter, and caecum and rectum weight. As far as we know, the effect of simultaneous dietary supplementation with lysine-methionine and L-carnitine on intestine segments is not documented. Some studies underline the relationship to villi surface area to better feed utilization, higher nutrient absorption, body weight gain, and growth performance [59,60]. However, Saki et al. [61] reported that the dietary addition of 0.36% of methionine in broilers feed did not affect intestinal villi characteristics on the 21^th^ and 42^nd^ days of age.

### 4.4. Humoral Immune Response

IgM primary response against the SRBC of birds fed with supplemented diets with L-carnitine and lysine-methionine was significantly lower than that of birds receiving the unsupplemented diet. No significant differences have been observed among dietary treatments for IgT and IgG titers against SRBC during both primary and secondary antibody responses. Latshaw [62] reported that antibodies are proteins; therefore, any deficiency of essential amino acids, particularly during the growth of chickens, results in poor immune competence. Lysine is one of the amino acids that can influence the magnitude of antibody response [63,64]. This could be the reason that the lowest immune response was observed in the control diet where lysine was not supplemented. Therefore, a 15% of lysine broiler chicken diet was sufficient to stimulate optimum antibody production; therefore, a lower immune response was observed when lysine was added at 30% and 75% in excess of the NRC.

Reports on the effect of methionine supplementation on broiler chickens’ humoral immune response are lacking. However, as far as L-carnitine was concerned, Deng et al. [65] found that the dietary addition of 0 (control), 100 mg/kg, or 1000 mg/kg of L-carnitine did not affect primary antibody responses to SRBC at the 4^th^ week but that birds fed on a diet with 1000 mg of L-carnitine had a higher primary antibody response against SRBC than broilers in other groups at the 12^th^ week. Moghaddam and Emadi [66] reported that there was a tendency for an increase in IgG, IgM, and IgA antibody titers as dietary threonine increased from 0.8% to 0.87%. However, IgG, IgM, and IgA antibody titers decreased when threonine was administrated at levels of 0.94% and 1.01%. The titers of IgG, IgM, and IgA antibodies for a secondary response were higher than those for a primary response.

## 5. Conclusions

The findings of this study suggest that the dietary supplementation of L-carnitine solely or in combination with excesses of lysine or methionine (with respect to NRC recommendations) did not affect body weight gain, feed conversion, and economical carcass segments of broiler chickens. By contrast, the diet with around 30% of lysine-methionine content increased the back thoracic vertebrae and the proventriculus weights. A combination of lysine-methionine (level equal to NRC recommendations) with L-carnitine (15% and 75%) improved the immune response of broiler chickens against Newcastle and Gumboro diseases by stimulating the antibody production.

## Figures and Tables

**Table 1 animals-09-00362-t001:** Ingredients and chemical compositions of the control diet.

Ingredient (%)	Starter Period (Day 1–22 of Age)	Grower Period (Day 23–35)	Finisher Period (Day 36–42)
corn	59.0	62.0	65.0
soybean meal	35.0	30.0	27.0
soybean oil	2.0	3.5	3.5
Ca22% P18%	1.5	1.8	1.7
mineral oysters	1.2	1.5	1.5
NaCl	0.3	0.2	0.2
lysine-hydro-chloride	0.1	0.1	0.1
DL-methionine	0.2	0.2	0.2
mineral premix *	0.3	0.3	0.3
vitamin premix **	0.3	0.3	0.3
Sodium hydrogen carbonate (NaHCO_3_)	0.1	0.1	0.1
**Chemical Composition (%)**			
metabolizable energy (kcal/kg)	2898	2955	3058
crude protein	21.7	20.8	18.2
crude fiber	2.3	2.2	2.1
calcium	0.80	0.80	0.79
available potassium	0.48	0.46	0.42
sodium	0.15	0.15	0.15
lysine	1.41	1.26	1.22
methionine	0.61	0.57	0.48
energy: protein ratio	133	142	167
Ca: P	1.7	1.7	1.9
linoleic acid	2.46	2.73	2.95
methionine + cysteine	0.97	0.91	0.77

***** Calcium Pantothenate: 4 mg/g; Niacin: 15 mg/g; Vitamin B6: 13 mg/g; Cu: 3 mg/g; Zn: 15 mg/g; Mn: 20 mg/g; Fe: 10 mg/g; K: 0.3 mg/g; ****** Vitamin A: 5000 IU/g; Vitamin D3: 500 IU/g; Vitamin E: 3 mg/g; Vitamin K3: 1.5 mg/g; Vitamin B2: 1 mg/g.

**Table 2 animals-09-00362-t002:** Amount of L-carnitine, lysine and methionine in nutrient analysis of studied treatments.

Variation	Diets.								
	C	D1	D2	D3	D4	D5	D6	D7	D8
**L-carnitine ***	NRC	NRC	NRC	NRC + 15%	NRC + 15%	NRC + 15%	NRC + 75%	NRC + 75%	NRC + 75%
**Lysine ****	NRC	NRC + 15%	NRC + 30%	NRC	NRC + 15%	NRC + 30%	NRC	NRC + 15%	NRC + 30%
**Methionine *****	NRC	NRC + 15%	NRC + 30%	NRC	NRC + 15%	NRC + 30%	NRC	NRC + 15%	NRC + 30%

* L-carnitine (NRC): Starter period: 17.8 mg/kg; Grower period: 18.1 mg/kg; Finisher period: 22.9 mg/kg; ** lysine (NRC): Starter period: 1.41%; Grower period: 1.26%; Finisher period: 1.22%; *** methionine (NRC): Starter period: 0.61%; Grower period: 0.57%; Finisher period: 0.48%

**Table 3 animals-09-00362-t003:** Growth performance of Ross 308 broilers fed diets containing different levels of L-carnitine and lysine-methionine from day 1 to day 42 of age.

Parameters	Period	Diets									SEM	*p*
		C	D1	D2	D3	D4	D5	D6	D7	D8		
Feed intake, g/pen/period	1–21 d	997.30	1012.60	1009.90	988.67	977.43	983.57	1005.43	1012.30	1006.53	8.37	NS
	22–35 d	2815.30	2898.46	2892.30	2819.50	2810.20	2800.90	2814.60	2859.19	2793.13	50.49	NS
	35–42 d	1851.90 ^a^	1702.40 ^ab^	1781.90 ^ab^	1862.00 ^a^	1748.9 ^ab^	1564.40 ^b^	1945.10 ^a^	1872.80 ^a^	1757.70 ^ab^	83.30	*
	1–42 d	5664.50 ^a^	5607.46 ^ab^	5684.10 ^ab^	5670.17 ^ab^	5536.53 ^ab^	5348.87 ^b^	5765.13 ^a^	5744.29 ^a^	5557.36 ^a^	100.06	*
Body weight gain, g/pen/period	1–21 d	697.60 ^ab^	706.31 ^a^	745.50 ^a^	671.10 ^abc^	621.60 ^bc^	610.17 ^c^	733.67 ^a^	741.00 ^a^	723.27 ^a^	18.57	**
	22–35 d	1207.10 ^a^	1241.29 ^a^	1165.50 ^ab^	1066.67 ^c^	1029.83 ^c^	1042.00 ^c^	1115.17 ^bc^	1100.41 ^bc^	1089.90 ^bc^	29.31	**
	35–42 d	1935.60 ^a^	1898.60 ^a^	1917.60 ^a^	1889.03 ^a^	1949.27 ^a^	1835.33 ^a^	1037.36 ^b^	1010.89 ^b^	1976.53 ^a^	69.53	*
	1–42 d	2840.30 ^a^	2846.20 ^a^	2828.60 ^a^	2626.80 ^ab^	3600.70 ^ab^	3602.70 ^b^	2886.20 ^a^	2852.30 ^a^	2789.70 ^a^	86.91	*
Feed Conversion ratio	1–21 d	1.43 ^bc^	1.43 ^bc^	1.35 ^c^	1.47 ^bc^	1.57 ^ab^	1.61 ^a^	1.37 ^c^	1.36 ^c^	1.39 ^c^	0.03	**
	22–35 d	2.33 ^c^	2.33 ^c^	2.48 ^bc^	2.64 ^ab^	2.72 ^a^	2.69 ^a^	2.52 ^ab^	2.60 ^ab^	2.56 ^ab^	0.06	**
	35–42 d	1.98 ^a^	1.89 ^a^	2.00 ^a^	2.09 ^a^	1.84 ^a^	1.87 ^a^	1.88 ^a^	1.86 ^a^	1.82 ^a^	0.09	NS
	1–42 d	1.99 ^ab^	1.97 ^b^	2.02 ^ab^	2.15 ^a^	2.12 ^ab^	2.15 ^a^	1.99 ^ab^	2.01 ^ab^	1.99 ^ab^	0.04	*

C (Control) = diet with lysine, methionine, and L-carnitine equal to NRC recommendations; D1 = control diet supplemented with lysine at 15% in excess of NRC, methionine at 15% in excess of NRC, and L-carnitine equal to NRC; D2 = control diet supplemented with lysine at 30% in excess of NRC, at 30% in excess of NRC, and L-carnitine equal to NRC; D3 = control diet supplemented with lysine equal to NRC, methionine equal to NRC, and L-carnitine at 15% % in excess of NRC; D4 = control diet supplemented control diet supplemented with lysine at 15% in excess of NRC, methionine at 15% in excess of NRC, and L-carnitine at 15%in excess of NRC; D5 = control diet supplemented lysine at 30% in excess of NRC, methionine at 30% in excess of NRC, and L-carnitine at 15% in excess of NRC; D6 = control diet supplemented with lysine equal to NRC recommendations, methionine equal to NRC recommendations, and L-carnitine at 75% in excess of NRC; D7 = control diet supplemented with lysine at 15% in excess of NRC, methionine at 15% in excess of NRC, and L-carnitine at 75% in excess of NRC; D8 = control diet supplemented with lysine at 30% in excess of NRC, methionine at 30% in excess of NRC, and L-carnitine at 75% in excess of NRC; SEM = standard error of the mean; ** *p* < 0.001, * *p* < 0.05, NS = *p* ≥ 0.05; a, b: Means within the same row without common superscripts letters are not significantly different (*p* ≥ 0.05).

**Table 4 animals-09-00362-t004:** Economical carcass segment means of Ross 308 broilers fed diets containing different levels of L-carnitine, lysine, and methionine from day 1 to day 42 of age.

Traits		Diets									SEM	*p*
		C	D1	D2	D3	D4	D5	D6	D7	D8		
**Economical carcass segments**	**Live body weight, g**	2873.70	2746.00	2826.00	2894.70	2883.30	2530.70	2600.40	2638.70	2523.30	172.46	NS
	**Full abdomen carcass weight, g**	2737.70	2619.00	2682.70	2738.30	2657.70	2356.00	2466.70	2487.30	2374.00	168.39	NS
	**Breast weight, g**	789.51	758.60	679.33	733.33	719.33	643.33	657.33	625.33	619.33	61.26	NS
	**Drumsticks (thighs) weight, g**	651.00 ^a^	583.00 ^ab^	520.67 ^b^	560.67 ^ab^	565.33 ^ab^	496.67 ^b^	514.670 ^b^	502.00 ^b^	508.00 ^b^	37.72	*
**Body organ segments**	**Neck weight, g**	79.57 ^a^	64.50 ^ab^	61.46 ^b^	58.66 ^b^	59.37 ^b^	51.88 ^b^	59.97 ^b^	60.20 ^b^	64.63 ^ab^	5.22	*
	**Back thoracic vertebrae (notarium) weight, g**	45.45 ^b^	64.28 ^ab^	103.99 ^a^	82.34 ^ab^	82.93 ^ab^	85.18 ^ab^	76.65 ^ab^	78.41 ^ab^	45.11 ^b^	12.92	*
	**Heart weight, g**	17.19	16.52	17.24	16.39	18.23	15.17	16.21	15.66	15.40	1.51	NS
	**Liver weight, g**	70.13	64.20	72.31	72.40	69.66	58.38	61.27	65.60	59.97	4.82	NS
	**Gizzard (ventriculus) weight, g**	65.37	50.43	62.47	65.78	62.00	53.19	52.23	58.96	55.05	5.52	NS
	**Abdominal fat weight, g**	43.95	37.96	41.07	25.57	38.33	31.75	32.47	41.16	24.43	6.86	NS
**Gut organs**	**Pancreas weight, g**	6.07	6.77	6.62	6.84	7.01	7.37	6.65	6.58	5.71	0.61	NS
	**Crop weight, g**	8.63	7.38	7.47	10.10	7.58	7.35	7.23	7.90	6.98	0.99	NS
	**Proventriculus weight, g**	10.77 ^ab^	8.59 ^b^	11.26 ^a^	9.15 ^ab^	11.43 ^a^	10.37 ^ab^	10.91 ^ab^	10.60 ^ab^	9.13 ^ab^	0.72	NS

C (Control) = diet with lysine, methionine, and L-carnitine equal to NRC recommendations; D1 = control diet supplemented with lysine at 15% in excess of NRC, methionine at 15% in excess of NRC, and L-carnitine equal to NRC; D2 = control diet supplemented with lysine at 30% in excess of NRC, methionine at 30% in excess of NRC, and L-carnitine equal to NRC; D3 = control diet supplemented with lysine equal to NRC, methionine equal to NRC, and L-carnitine at 15% in excess of NRC; D4 = control diet supplemented control diet supplemented with lysine at 15% in excess of NRC, methionine at 15% in excess of NRC, and L-carnitine at 15% in excess of NRC; D5 = control diet supplemented lysine at 30% in excess of NRC, methionine at 30% in excess of NRC, and L-carnitine at 15% in excess of NRC; D6 = control diet supplemented with lysine equal to NRC recommendations, methionine equal to NRC recommendations, and L-carnitine at 75% in excess of NRC; D7 = control diet supplemented with lysine at 15% in excess of NRC, methionine at 15% in excess of NRC, and L-carnitine at 75% in excess of NRC; D8 = control diet supplemented with lysine at 30% in excess of NRC, methionine at 30% in excess of NRC, and L-carnitine at 75% in excess of NRC; SEM = standard error of the mean; * *p* < 0.05; NS = *p* ≥ 0.05; a,b: Means within the same row without common superscripts letters are not significantly different (*p* ≥ 0.05).

**Table 5 animals-09-00362-t005:** Ross 308 broilers’ intestine segment dimensions according to feed diets from 1–6 weeks of age.

Traits		Diets									SEM	*p*
		C	D1	D2	D3	D4	D5	D6	D7	D8		
**Duodenum**	**Weight, g**	20.27 ^a^	15.53 ^ab^	14.90 ^ab^	16.37 ^ab^	15.64 ^ab^	14.61 ^ab^	13.56 ^b^	14.02 ^b^	12.99 ^b^	1.77	*
	**Length, mm**	34.33 ^a^	33.00 ^ab^	29.60 ^abc^	30.30 ^abc^	26.60 ^bcd^	27.60 ^abcd^	24.00 ^cd^	22.00 ^d^	25.00 ^cd^	2.27	*
	**Width, mm**	7.50 ^b^	7.97 ^ab^	7.81 ^b^	8.29 ^ab^	7.64 ^b^	6.84 ^b^	8.22 ^ab^	8.13 ^ab^	9.38 ^a^	0.57	*
	**Diameter, mm**	0.94 ^b^	1.16 ^b^	1.17 ^b^	1.61 ^a^	1.67 ^a^	1.68 ^a^	0.89 ^b^	1.05 ^b^	1.21 ^b^	0.12	*
**Jejunum**	**Weight, g**	60.09	51.16	51.07	57.91	52.52	47.75	46.30	60.88	46.84	0.70	NS
	**Length, mm**	120.00	113.10	112.30	110.00	111.66	108.33	109.00	113.66	101.00	5.71	NS
	**Width, mm**	9.26 ^abc^	8.44 ^bc^	8.23 ^a^	9.50 ^ab^	9.80 ^a^	9.34 ^abc^	8.26 ^bc^	8.02 ^c^	9.30 ^abc^	0.40	*
	**Diameter, mm**	1.26 ^a^	1.40 ^abc^	1.01 ^c^	1.25 ^bc^	1.34 ^bc^	1.59 ^ab^	1.09 ^bc^	1.20 ^bc^	1.80 ^a^	0.15	*
**Ileum**	**Weight, g**	9.10 ^a^	5.23 ^bc^	5.06 ^bc^	6.39 ^b^	2.67 ^d^	3.95 ^dc^	3.39 ^dc^	2.97 ^dc^	3.09 ^dc^	4.82	**
	**Length, mm**	18.16 ^a^	16.80 ^ab^	11.16 ^cd^	15.23 ^abc^	11.66 ^cd^	12.66 ^bc^	13.00 ^bcd^	8.00 ^d^	9.26 ^d^	1.51	**
	**Width, mm**	7.33 ^ab^	5.97 ^ab^	7.26 ^ab^	5.39 ^b^	5.27 ^b^	5.92 ^ab^	6.45 ^ab^	7.97 ^a^	6.60^ab^	0.61	*
	**Diameter, mm**	1.03	1.14	1.05	1.43	1.48	1.54	1.35	1.08	1.32	0.16	NS
**Rectum weight, g**		2.14	1.80	2.29	2.23	2.11	2.30	1.78	2.32	1.82	0.30	NS
**Cecum weight, g**		17.13	14.22	14.12	12.74	14.54	12.62	13.74	15.97	14.56	1.98	NS
**Colon weight, g**		2.10 ^ab^	1.28 ^b^	1.73 ^ab^	1.52 ^ab^	2.36 ^a^	1.87 ^ab^	1.98 ^ab^	1.62 ^ab^	2.06 ^ab^	0.26	*

C (Control) = diet with lysine, methionine, and L-carnitine equal to NRC recommendations; D1 = control diet supplemented with lysine at 15% in excess of NRC, methionine at 15% in excess of NRC, and L-carnitine equal to NRC; D2 = control diet supplemented with lysine at 30% in excess of NRC, methionine at 30% in excess of NRC, and L-carnitine equal to NRC; D3 = control diet supplemented with lysine equal to NRC, methionine equal to NRC, and L-carnitine at 15% in excess of NRC; D4 = control diet supplemented control diet supplemented with lysine at 15% in excess of NRC, methionine at 15% in excess of NRC, and L-carnitine at 15% in excess of NRC; D5 = control diet supplemented lysine at 30% in excess of NRC, methionine at 30% in excess of NRC, and L-carnitine at 15% in excess of NRC; D6 = control diet supplemented with lysine equal to NRC recommendations, methionine equal to NRC recommendations, and L-carnitine at 75% in excess of NRC; D7 = control diet supplemented with lysine at 15% in excess of NRC, methionine at 15% in excess of NRC, and L-carnitine at 75% in excess of NRC; D8 = control diet supplemented with lysine at 30% in excess of NRC, methionine at 30% in excess of NRC, and L-carnitine at 75% in excess of NRC, SEM = standard error of the mean; ** *p* < 0.001, * *p* < 0.05; NS = *p* ≥ 0.05; a,b: Means within the same row without common superscripts letters are not significantly different (*p* ≥ 0.05).

**Table 6 animals-09-00362-t006:** Anti- Sheep Red Blood Cells (SRBC) antibody responses (log_10_) of broilers fed different levels of L-carnitine and lysine-methionine.

Dietary Treatment	Primary Response			Secondary Response		
	Total Antibody	IgM	IgG	Total Antibody	IgM	IgG
**C**	4.66	2.33 ^a^	2.33	6.66	3.66	3.00
**D1**	4.00	2.00 ^ab^	2.00	8.66	3.00	5.66
**D2**	3.33	1.66 ^ab^	1.66	5.66	2.33	3.33
**D3**	2.66	1.33 ^ab^	1.33	5.33	2.66	2.66
**D4**	3.33	1.33 ^ab^	2.00	7.00	3.00	4.00
**D5**	2.33	1.00 ^b^	1.33	6.33	2.66	3.66
**D6**	2.66	1.33 ^ab^	1.33	6.33	3.00	3.33
**D7**	2.66	1.33 ^ab^	1.33	7.00	3.00	4.00
**D8**	3.33	2.00 ^ab^	1.33	8.00	3.00	5.00
**SEM**	0.80	0.38	0.58	1.03	0.52	0.95
***p***	NS	*	NS	NS	NS	NS

C (Control) = diet with lysine, methionine, and L-carnitine equal to NRC recommendations; D1 = control diet supplemented with lysine at 15% in excess of NRC, methionine at 15% in excess of NRC, and L-carnitine equal to NRC; D2 = control diet supplemented with lysine at 30% in excess of NRC, methionine at 30% in excess of NRC, and L-carnitine equal to NRC; D3 = control diet supplemented with lysine equal to NRC, methionine equal to NRC, and L-carnitine at 15% in excess of NRC; D4 = control diet supplemented control diet supplemented with lysine at 15% in excess of NRC, methionine at 15% in excess of NRC, and L-carnitine at 15% in excess of NRC; D5 = control diet supplemented lysine at 30% in excess of NRC, methionine at 30% in excess of NRC, and L-carnitine at 15% in excess of NRC; D6 = control diet supplemented with lysine equal to NRC recommendations, methionine equal to NRC recommendations, and L-carnitine at 75% in excess of NRC; D7 = control diet supplemented with lysine at 15% in excess of NRC, methionine at 15% in excess of NRC, and L-carnitine at 75% in excess of NRC; D8 = control diet supplemented with lysine at 30% in excess of NRC, methionine at 30% in excess of NRC, and L-carnitine at 75% in excess of NRC, SEM = standard error of the mean; * *p* < 0.05, NS = *p* ≥ 0.05; a,b: Means within the same row without common superscripts letters are not significantly different (*p* ≥ 0.05).

**Table 7 animals-09-00362-t007:** Bronchitis, Newcastle, and Gumboro hemagglutination-inhibition (log_10_) titers of broilers fed on different levels of L-carnitine and lysine-methionine.

Traits	Diets									SEM	*p*
	C	D1	D2	D3	D4	D5	D6	D7	D8		
Antibody titer against injection of bronchitis at 23rd day of age	3.29	2.91	3.02	3.29	3.09	3.49	3.50	3.34	2.89	0.81	NS
Antibody titer against injection of Newcastle at 27th day of age (lg2)	3.66 ^b^	5.00 ^ab^	4.33 ^ab^	6.00 ^a^	4.66 ^ab^	5.66 ^ab^	4.66 ^ab^	4.00 ^ab^	4.66 ^ab^	0.60	*
Antibody titer against injection of Gumboro at 30th day of age	3.38 ^bc^	3.16^c^	3.57 ^ab^	3.51 ^ab^	3.44 ^abc^	3.54^ab^	3.59 ^a^	3.46 ^ab^	3.44 ^abc^	0.41	*
Bursa of fabricius weight, g	4.57	4.21	3.67	2.99	2.45	2.56	2.45	2.10	3.26	0.77	NS
Thymus weight, g	15.59 ^bc^	19.00 ^ab^	24.60 ^a^	15.39 ^bc^	7.17 ^d^	11.50 ^cd^	19.04 ^ab^	15.50 ^bc^	13.98 ^bc^	1.89	**
Spleen weight, g	3.61	3.04	2.94	2.86	2.47	2.61	2.92	2.90	3.00	0.40	NS

C (Control diet) with lysine, methionine, and L-carnitine equal to NRC recommendations; D1 = control diet supplemented with lysine at 15% in excess of NRC, methionine at 15% in excess of NRC, and L-carnitine equal to NRC; D2 = control diet supplemented with lysine at 30% in excess of NRC, methionine at 30% in excess of NRC, and L-carnitine equal to NRC; D3 = control diet supplemented with lysine equal to NRC, methionine equal to NRC, and L-carnitine at 15% in excess of NRC; D4 = control diet supplemented control diet supplemented with lysine at 15% in excess of NRC, methionine at 15% in excess of NRC, and L-carnitine at 15% in excess of NRC; D5 = control diet supplemented lysine at 30% in excess of NRC, methionine at 30% in excess of NRC, and L-carnitine at 15% in excess of NRC; D6 = control diet supplemented with lysine equal to NRC recommendations, methionine equal to NRC recommendations, and L-carnitine at 75% in excess of NRC; D7 = control diet supplemented with lysine at 15% in excess of NRC, methionine at 15% in excess of NRC, and L-carnitine at 75% in excess of NRC; D8 = control diet supplemented with lysine at 30% in excess of NRC, methionine at 30% in excess of NRC, and L-carnitine at 75% in excess of NRC; SEM = standard error of the mean; ** *p* < 0.001, * *p* < 0.05, NS = *p* ≥ 0.05; a,b: Means within the same row without common superscripts letters are not significantly different (*p* ≥ 0.05).

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
