# Peer review of "Effects of Dietary Supplementation of L-Carnitine and Excess Lysine-Methionine on Growth Performance, Carcass Characteristics, and Immunity Markers of Broiler Chicken"

_animals, 2019, doi:10.3390/ani9060362_

Round 1

Reviewer 1 Report

Dear Authors,

thank you for submitting your research results. Unfortunately, your manuscript need significant improvements in content, presentation and, spelling as well as formatting.

The abstract is not sufficient. You are presenting some results just by labelling the group name without explaining the differences of the groups. Also the alert reader does not know which parameters are included by writing "all parameters".

The introduction must be extended. Please focus more on chicken because birds are a broad class with many different species (e. g. there are huge differences between a cat and a cow). Please also add some references, e. g. in line 70. Also writings like "to the best knowledge" must be changed into "to the knowledge". But even then, that is no excuse for a lack of literature research.

Please be consistent in your abbreviations once you introduce them. NC (line 43) vs NDC (line 143) or C (line 90) vs. CD (table 3) are just two of the many examples which makes reading uneasy and irritates.

You should also introduce the abbreviation before using them (line 153).

In table 1, the data regarding L-carnitine is missing, nevertheless you are referring to this data several times.

The methods are not described properly. The alert reader has no idea about the selection process of chicken. 3 groups with each 10 chicks were fed one diet. Are you selecting one chicken per same-diet-pens (n=3) for the slaughter or three chicken of one pen and none of the other two pens. Please rewrite the whole method section to make everything understandable and clear.

Regarding the labelling of the groups, I do not understand why D6, D7 and D8 are no named D3, D4, D5. It makes more sense to grade the groups also by increased supplementation of L-carnitine.

The result section, as well as the discussion section and conclusion section must be rewritten too. The presentation of the results is not acceptable. For example. you are referring to results in table 4 before even introducing this table. The data in the tables are also not formatted correctly and homogenously (112.30 vs 112.3 vs 112). Please also recount the data because the sums are not correct quite often.

After rewriting the paper and taking more care in details, I would be pleased to review it again. But unfortunately, I have to reject the submitted manuscript.

Kind regards

Reviewer 2 Report

In this manuscript "Dietary supplementation of L-Carnitine and lysine-methionine excess on growth performance, carcass characteristics and immunity markers of broiler chicken” by Seyed Mohammad Ghoreyshi and co-authors presents the results of their studies on effect of feed supplementation with carnitine, lysine-methionine on poultry nutrition. The present results confirmed that the amino acid supplementation in poultry feed enhances the growth, and immunity of chicken.

There are a few points that have to be addressed and rewrite the manuscripts before publishing: Major revision

1.     Title: Authors should modify title as – Effects of dietary supplementation of L-Carnitine and excess lysine-methionine on growth performance, carcass characteristics and immunity markers of broiler chicken.

2.     It would be better if the authors provide clear picture of 8 different diets in abstract—what is there in those 8 diets and how they are different with each other.

3.     In page2, line 55: please check the spelling of insulin.

4.     Authors are suggested to use superscript option for indicating correct expression of days, electron valence of chemicals throughout the manuscript. eg: Fe2+ instead of Fe2+, page 4, line 126: 1st, 21st, 22nd, 35th instead of 1st, 21st, 22nd, 35th.     

5.     Animal ethics approval number should be provided.

6.     Page 3, line 99: Check the word ‘Table 2’ is in middle. Authors are suggested to format it.

7.     Page 4, line 116: Include punctuation mark (comma [,]) after Two hundred seventy, 1-day-old.

8.     Why only male birds were used for the experiments?

9.     Authors are suggested to include few references at section no. 2.4 and 2.5 in page no. 4, and also recommend including reference “AK Panda et al., 2010. Asian-Australasian Journal of Animal Sciences 23 (12), 1626-1631”, “AK Panda et al., 2015. Indian Journal of Animal Sciences 85 (12), 1354-1357”, and “AK Panda et al., 2016. Animal Nutrition and Feed Technology 16 (3), 417-425”.

10.   Authors are suggested to strengthen the importance and relevance of the present results in the discussion section. Ex: Include the role of carnitine, lysine, and methionine in the discussion part.

11.  Authors should check the conclusion part: the results are really supporting the conclusion? The tested /selected parameters are sufficient for the conclusion?

12.  References should be cited by following journal style/format. References -9, 15, 16, 17, 26, 28, 30 are not in the journal format.

13.   Need to check for typographical errors, plagiarism, punctuation, and grammar throughout the manuscript.

Reviewer 3 Report

Title: Dietary supplementation of L-carnitine and 2 lysine-methionine excess on growth performance, 3 carcass characteristics and immunity markers of 4 broiler chicken.

Comments:

Introduction:

Line 68-70: add references to this paragraph.

Line 72: change Fe2+ by Fe2+.

Add a hypothesis at the end of the introduction.

Methods:

Line 88: indicate the bioethics certificate number.

Why the authors used the NRC requirements? if the Ross 308 broiler has a catalog with other requirements, which are higher than those of NRC.

Indicate the origin of the amino acids used and L-carnitine (name of supplier, country, etc.)

Lines 106-114: Delete the paragraph, with table 2 the experimental groups are well explained.

It is necessary to add proximal chemical analysis of the final diets (in table 2).

The experimental unit for live weight was each animal? explain why it's not clear.

Line 129: change par by per.

The N used for carcass characteristics is very low. The authors must justify it.

Indicate more details of the sacrifice of the animals.

Results and discussion

Review the rows in Table 1 (one line of BW appears in FCR).

Check that all values have the same decimals in the tables.

Lines 384-386: add some biological or physiological explanation of the observed results.

In general, the discussion of the article is poor and only based on comparisons with other works. It is necessary to discuss the results with a deeper physiological basis. This occurs in the growth performance, carcass characteristic, intestine segments and humoral response sections.

Add to the discussion what effects on health and nutrition could have the changes observed in the segments of the intestine, for example.

References

The list of references are not in the format of the journal.

For example, the name of the journal should be abbreviated (lines: 447, 449, 476, 479, and others).

The names of the articles should start with a capital letter and then with a lower case (line: 549).

Round 2

Reviewer 1 Report

Dear authors,

thank you very much for the rewritten manuscript. Unfortunately, you have not addressed some of my previous revisions.

Please be consistent in using capital or small letters (L-carnithine vs. L-Carnithine, methionine vs. Methionine, lysine vs. Lysine). There are several changes within the manuscript and title.

Line 35: Please do not use an abbreviation without explaining / introducing it (NRC)

Line 53: Please do not use an abbreviation without explaining / introducing it (SRBC)

Line 56: Please do not use an abbreviation without explaining / introducing it (ND)

Line 78:. Please focus more on chicken because birds are a broad class with many different species (e. g. there are huge differences between a cat and a cow). Please also add some references, e. g. in line 70. Also writings like "to the best knowledge" must be changed into "to the knowledge". But even then, that is no excuse for a lack of literature research.

Please be consistent in using the whole word or the introduced abbreviation (Line 74 Met and Lys, line 78 methionine and lysine etc.) There are several changes within the manuscript.

Line 103: Please chance "to the best knowledge" into "to the knowledge".

Line 101: In table 1, the data regarding L-carnitine (NRC) is missing, nevertheless you are referring to this data several times.

Line 101: Please use the same format for all data (e. g. two decimal places)

Line 134: You have an increased amount of Lys and Met (C-D1-D2, D3-D4-D5, D6-D7-D8) Why do you not order the three blocs (C-D1-D2, D3-D4-D5, D6-D7-D8) regarding the amount of L-carnithine (current order: NCR, NRC + 75%, NRC + 15%, more logical order: NCR, NRC + 15%, NRC + 75%)

Line 176: Please do not use an abbreviation without explaining / introducing it (GLM)

Line 189: The table is incorrect. Please check all numbers and sums (e.g. 997.3 + 2815.30 + 1851.90 is not 56640 but 5664.5) in the table as well as in the text. I do not want to check all numbers by myself but such a mistake is not appropriate

Line 307: Please do not use an abbreviation without explaining / introducing it (IBD)

Line 565 f.: Please check the format

Author Response

The authors want to express their gratitude to the reviewer for his/her valuable comments and suggestions. The authors’ replies to the individual points raised are reported in Italic below.

Open Review

English language and style

( ) Extensive editing of English language and style required 
( ) Moderate English changes required 
(x) English language and style are fine/minor spell check required 
( ) I don't feel qualified to judge about the English language and style 

Yes

Can be improved

Must be improved

Not applicable

Does the   introduction provide sufficient background and include all relevant   references?

(x)

( )

( )

( )

Is the research   design appropriate?

(x)

( )

( )

( )

Are the methods   adequately described?

(x)

( )

( )

( )

Are the results   clearly presented?

( )

( )

(x)

( )

Are the   conclusions supported by the results?

( )

(x)

( )

( )

Comments and Suggestions for Authors

Dear authors,

thank you very much for the rewritten manuscript. Unfortunately, you have not addressed some of my previous revisions.

Please be consistent in using capital or small letters (L-carnithine vs. L-Carnithine, methionine vs. Methionine, lysine vs. Lysine). There are several changes within the manuscript and title.

We have checked the paper in order to be consistent in using capital or small letters.

Line 35: Please do not use an abbreviation without explaining / introducing it (NRC)

Also in the Abstract this abbreviation was introduced.

Line 53: Please do not use an abbreviation without explaining / introducing it (SRBC)

Also in the Abstract the abbreviation was introduced.

Line 56: Please do not use an abbreviation without explaining / introducing it (ND)

This abbreviation was changed into NCD and it was introduced.

Line 78:. Please focus more on chicken because birds are a broad class with many different species (e. g. there are huge differences between a cat and a cow). Please also add some references, e. g. in line 70. Also writings like "to the best knowledge" must be changed into "to the knowledge". But even then, that is no excuse for a lack of literature research.

Additional and more recent references were added, also focusing on chicken.

Please be consistent in using the whole word or the introduced abbreviation (Line 74 Met and Lys, line 78 methionine and lysine etc.) There are several changes within the manuscript.

We have used the whole word along the manuscript.

Line 103: Please chance "to the best knowledge" into "to the knowledge".

“to the best knowledge” was replaced to  “to the knowledge”.

Line 101: In table 1, the data regarding L-carnitine (NRC) is missing, nevertheless you are referring to this data several times.

Done. Please see Table 2.

Line 101: Please use the same format for all data (e. g. two decimal places)

We have used data with two decimal places in the Tables and related text, excepted for some data of ingredients/nutrients in Table 1 where we prefer to use one decimal place or no decimal place for energy values.

Line 134: You have an increased amount of Lys and Met (C-D1-D2, D3-D4-D5, D6-D7-D8) Why do you not order the three blocs (C-D1-D2, D3-D4-D5, D6-D7-D8) regarding the amount of L-carnithine (current order: NCR, NRC + 75%, NRC + 15%, more logical order: NCR, NRC + 15%, NRC + 75%)

As you suggested, this remark was taken into consideration and order was checked.

Line 176: Please do not use an abbreviation without explaining / introducing it (GLM)

This abbreviation was introduced.

Line 189: The table is incorrect. Please check all numbers and sums (e.g. 997.3 + 2815.30 + 1851.90 is not 56640 but 5664.5) in the table as well as in the text. I do not want to check all numbers by myself but such a mistake is not appropriate

Done. This remark was taken into consideration and all values were checked.

Line 307: Please do not use an abbreviation without explaining / introducing it (IBD)

The abbreviation was replaced by IB, introduced in the previous text

Line 565 f.: Please check the format

The reference format was checked

Submission Date

18 March 2019

Date of this review

13 May 2019 12:26:50

Reviewer 2 Report

Authors have now improved the manuscript but still there are some concerns.

Table 4. Line number 231: There is lot of space between 15% in and excess, remove the space.

Still the authors are feeling difficult in arranging the references in journal format. Authors are suggested to use reference manager software to arrange the references uniformly as instructed in the author instructions of the journal.

Need to check for typographical errors, plagiarism, punctuation, and grammar throughout the manuscript.

Author Response

The authors want to express their gratitude to the reviewer for his/her valuable comments and suggestions. The authors’ replies to the individual points raised are reported in Italic below.

Inizio modulo

Review Report Form

Open Review

English language and style

( ) Extensive editing of English language and style required 
(x) Moderate English changes required 
( ) English language and style are fine/minor spell check required 
( ) I don't feel qualified to judge about the English language and style 

The linguistic revision of whole manuscript was carried out

Yes

Can be improved

Must be improved

Not applicable

Does the   introduction provide sufficient background and include all relevant   references?

(x)

( )

( )

( )

Is the   research design appropriate?

(x)

( )

( )

( )

Are the   methods adequately described?

(x)

( )

( )

( )

Are the   results clearly presented?

(x)

( )

( )

( )

Are the   conclusions supported by the results?

(x)

( )

( )

( )

Comments and Suggestions for Authors

Authors have now improved the manuscript but still there are some concerns.

Table 4. Line number 231: There is lot of space between 15% in and excess, remove the space.

The space was removed.

Still the authors are feeling difficult in arranging the references in journal format. Authors are suggested to use reference manager software to arrange the references uniformly as instructed in the author instructions of the journal.

A check of references format was carried out.

Need to check for typographical errors, plagiarism, punctuation, and grammar throughout the manuscript.

A check was carried out.

Submission Date

18 March 2019

Date of this review

14 May 2019 12:07:40

Fine modulo

© 1996-2019 MDPI (Basel, Switzer

Round 3

Reviewer 1 Report

Dear authors,

thank you very much for the improvements in the paper. There are serious flaws, which I adressed before and have not changed yet.

Table 3: All the data and sums within the table have not been checked and corrected (just the example I adressed in my previous review report). To be honest, I do not understand how you managed to count incorrectly. Besides the example of a mistake in this table, e. g. 745.50 + 1165.50 + 1917.60 ARE NOT 2828.60, BUT 3828.60. It is absolutely inadequate to find such mistakes anymore.

(By the way, why have you taken a different font size within the data?)

Table 2: I really appreciate that you have "We have checked the paper in order to be consistent in using capital or small letters" regarding L-carnithine vs. L-Carnithine, methionine vs. Methionine, lysine vs. Lysine. But during this process, you created "-Carnitinecarnitine", "Lysinelysine" and "Methioninemethionine" in the table and the description of the table.

Please check the whole paper and be more careful and precise.

I really hope that you can appreciate the effort reviewing manuscripts and put more attention into the corrections.

Kind regards

This manuscript is a resubmission of an earlier submission. The following is a list of the peer review reports and author responses from that submission.